# Validity of cerebrovascular ICD-9-CM codes in healthcare administrative databases. The Umbria Data-Value Project

**Massimiliano Orso**[1,2], **Francesco Cozzolino**[1,2], **Serena Amici**[3], **Marcello De Giorgi**[4], **David Franchini**[4], **Paolo Eusebi**[1], **Anna Julia Heymann**[5], **Guido Lombardo**[6], **Anna Mengoni**[2], **Alessandro Montedori**[1], **Giuseppe Ambrosio**[2], **Iosief Abraha**[1,7] *

**1** Health Planning Service, Regional Health Authority of Umbria, Perugia, Italy, **2** Division of Cardiology, Santa Maria della Misericordia Hospital, University of Perugia School of Medicine, Perugia, Italy, **3** Cognitive Disorder and Dementia Unit, USL Umbria, Perugia, Italy, **4** Health ICT Service, Regional Health Authority of Umbria, Perugia, Italy, **5** Istituto Zooprofilattico Sperimentale dell'Umbria e delle Marche, Perugia, Italy, **6** Department of Surgical and Biomedical Sciences, University of Perugia, Perugia, Italy, **7** Centro Regionale Sangue, Servizio Immunotrasfusionale, Azienda Ospedaliera di Perugia, Perugia, Italy

* iosief_a@yahoo.it

**Data Availability Statement:** All relevant data are within the manuscript and its Supporting Information files.

## Abstract

### Background

Validation of administrative databases for cerebrovascular diseases is crucial for epidemiological, outcome, and health services research. The aim of this study was to validate ICD-9 codes for hemorrhagic or ischemic stroke in administrative databases, to use them for a comprehensive assessment of the burden of disease in terms of major outcomes, such as mortality, hospital readmissions, and use of healthcare resources.

### Methods

We considered the hospital discharge abstract database of the Umbria Region (890,000 residents). Source population was represented by patients aged >18 discharged from hospital with a diagnosis of hemorrhagic or ischemic stroke between 2012 and 2014 using ICD-9-CM codes in primary position. We randomly selected and reviewed medical charts of cases and non-cases from hospitals. For case ascertainment we considered symptoms and instrumental tests reported in the medical charts. Diagnostic accuracy measures were computed using 2x2 tables.

### Results

We reviewed 767 medical charts for cases and 78 charts for non-cases. Diagnostic accuracy measures were: subarachnoid hemorrhage: sensitivity (SE) 100% (95% CI: 97%-100%), specificity (SP) 96% (90–99), positive predictive value (PPV) 98% (93–100), negative predictive value (NPV) 100% (95–100); intracerebral hemorrhage: SE 100% (97–100), SP 98% (91–100), PPV 98% (94–100), NPV 100% (95–100); other and unspecified intracranial hemorrhage: SE 100% (97–100), SP 96% (90–99), PPV 98% (93–100), NPV 100% (95–100); ischemic stroke due to occlusion and stenosis of precerebral arteries: SE 99%

**Funding:** This study was developed within the Data-Value Project ("Progetto Data-Value: valorizzazione del dato sanitario regionale per la Ricerca dei Servizi Sanitari (Health Services Research)" – D.G.R. No 1798 of 29/12/2014) supported by funding from the Regional Health Authority of Umbria. The funder had no role in study design, data collection and analysis, decision to publish, or preparation of the manuscript.

**Competing interests:** The authors have declared that no competing interests exist.

(94–100), SP 66 (57–75), PPV 70% (61–77), NPV 99% (93–100); occlusion of cerebral arteries: SE 100% (97–100), SP 87% (78–93), PPV 91% (84–95), NPV 100% (95–100); acute, but ill-defined, cerebrovascular disease: SE 100% (97–100), SP 78% (69–86), PPV % 83 (75–89), NPV 100% (95–100).

## Conclusions

Case ascertainment for both ischemic and hemorrhagic stroke showed good or high levels of accuracy within the regional healthcare databases in Umbria. This database can confidently be employed for epidemiological, outcome, and health services research related to any type of stroke.

## Introduction

In Italy, stroke is one of the leading cause of mortality, and the first cause of long-term disability. The overall Italian population is 60,656,000 and the incidence estimate is 73,116 strokes/year, the prevalence estimate is 351,820 strokes, with 75,252 deaths/year due to stroke [1]. Stroke represents a major social and economic problem: the healthcare estimated cost amounts to € 3,195.9 million, € 53 per capita. In Europe, we expect an increase of about 32% of incidence in the next twenty years, mainly due to the aging of the population [2]. Epidemiological data of stroke in Italy are in line with those of other high-income countries [3].

Trends in epidemiology and survival rates of stroke can be assessed using administrative healthcare databases. Administrative databases have the advantage that they can link different sources of information (such as discharge data, prescription and laboratory data) providing a comprehensive understanding of the burden of stroke in terms of important outcomes such as mortality, disability, hospital readmissions, and use of healthcare resources. However, these databases need to be adequately validated by comparing their main content, that is, the diagnosis represented by the International Classification of Diseases (ICD), with another source of information such as the clinical chart [4, 5]. In addition, such databases can aid in monitoring adherence to drug therapies, including the use of evidence-based therapies. For these reasons, the validation of available administrative databases for cerebrovascular diseases becomes crucial, and it represents the first step for the conduction of subsequent epidemiological, outcome, and health services research.

The objective of the present study was to evaluate the accuracy of the ICD-9-CM codes in identifying patients with hemorrhagic or ischemic stroke in the administrative database of the Regional Health Authority of Umbria.

## Materials and methods

### Setting and data source

**Administrative database.** The regional healthcare administrative database of Umbria gathers information regarding all hospital admission medical records on all 890,000 residents, including personal demographics, hospital admission and discharge dates, vital status, hospital department, primary and secondary diagnoses, and surgical or diagnostic procedures. In addition, the database records all drug prescriptions listed in the National Drug Formulary, and it allows identification of the prescriber. The regional administrative database has already been used for pharmacoepidemiology and drug-related outcome research [6–9]. Each resident has a

unique identification code within the database, to which the various types of information corresponding to each individual are linked. In Italy, healthcare is provided almost entirely by the Italian National Health System (NHS), therefore most residents' significant healthcare information can be retrieved within the healthcare databases.

**Source population.** Source population was represented by permanent Umbria Region residents aged 18 or above. Any resident that had been discharged (with exclusion of voluntary discharge and inter-hospital transfer of patients) from a hospital with a diagnosis of hemorrhagic or ischemic stroke was considered. Residents that have been hospitalised outside the regional territory of Umbria were excluded from analysis.

**Case selection and sampling method.** The research protocol for this study has been previously published [10]. From the discharge abstract database of Umbria we identified by a simple randomization method using SAS 9.4 procedures six cohorts of "cases", that is patients with a diagnosis of hemorrhagic or ischemic stroke, between 2012 and 2014 having the ICD-9 codes located in primary position of subarachnoid hemorrhage (ICD-9 code 430), intracerebral haemorrhage (code 431), other and unspecified intracranial hemorrhage (codes 432.x), occlusion and stenosis of paracerebral arteries (codes 433.x1), occlusion of cerebral arteries (codes 434.x1), and acute but ill-defined cerebrovascular disease (code 436).

We excluded prevalent cases, i.e. patients discharged from hospitals with any of the diagnostic codes investigated with the same diagnosis in the five years before the index date. From each cohort, we extracted a random sample of 130 cases (see Statistical Methods for details). At the same time, we identified a cohort of "non-cases", corresponding to patients who had been discharged in the same period with a diagnosis of cerebrovascular disease (ICD-9 codes 390–459), including transient ischemic attack, but without hemorrhagic or ischemic stroke. From this cohort of non-cases, we extracted a random sample of 80 patients. This sample of non-cases was used as control for each of the six conditions.

## Chart abstraction and case ascertainment

We examined medical charts of cases and non-cases from hospitals archives for case ascertainment.

We collected the following information from the medical charts: unique identification patient code, date of birth, gender, dates of hospital admission and discharge, any diagnostic procedure and treatment that contributed to the diagnosis of the disease.

The chart reviewers were physicians appropriately trained in data extraction. A pilot phase was performed in which they independently examined 30 medical charts. The level of agreement between the two reviewers was very high (k> 0.85). To ensure a higher level of agreement the results of the pilot phase were discussed by the working group. Disagreements were resolved through the involvement of a third reviewer (GA). Each reviewer independently completed the data extraction using standardized forms.

The detection of symptoms and diagnostic tests have been considered for the case ascertainment, as described below.

## Validation criteria

As defined in the study protocol [10], we used the criteria indicated in the published international guidelines regarding hemorrhagic and ischemic stroke to validate the related ICD-9-CM codes. To validate hemorrhagic stroke ICD-9 codes (430, 431, 432.x), we also considered the criteria defined by the American Heart Association/American Stroke Association (AHA/ASA)[11, 12], while to validate ischemic stroke ICD-9 codes (433.x1, 434.x1, 436) the criteria defined by the AHA/ASA[13] and European Stroke Organization (ESO) were used[14].

According to the above-mentioned guidelines, for the validation of hemorrhagic and ischemic stroke we considered the corresponding ICD-9 codes valid when both of the following conditions were present: (1) detection of focal lesions by neurological examination; (2) imaging test (CT, MRI, or angiography).

Neuroimaging was the main discriminator between the two types of stroke: the presence of hemorrhagic lesion classified the case as hemorrhagic stroke, while negative imaging for hemorrhage classified the case as ischemic stroke.

## Statistical analysis

As previously published, for each ICD-9 code we calculated a sample of 121 cases in order to obtain an expected sensitivity of 80%, with a half-width of the 95% CI equal to 8% [10]. For specificity, we calculated a sample of 73 non-cases (cerebrovascular disease patients without the diseases of interest) to obtain an expected specificity of 90%, with a half-width of the 95% CI equal to 8%, according to binomial exact calculation [15]. We considered published systematic reviews to derive the expected accuracy measures [4, 5]. In order to provide for potential missing medical charts, we decided to review a higher number of medical charts than anticipated.

For each ICD-9-CM code we calculated sensitivity and specificity, with their corresponding 95% CI. Sensitivity expressed the proportion of 'true positives' (i.e., patients with subarachnoid hemorrhage classified as positive by the administrative database and medical record review) relative to all cases deemed positive by the medical chart review. Specificity expressed the proportion of 'true negatives' (i.e., subarachnoid hemorrhage identified as negative by the administrative database and medical record review) relative to all cases deemed negative by the medical chart review. Positive and negative predicting values were also calculated, along with their 95% CI.

## Reporting

We ensured quality of reporting following the criteria of Standards for Reporting Diagnostic Accuracy (STARD) 2015 [16] (**S1 Table**).

## Ethics statement

Ethics approval has been obtained from the Regional Ethics Committee of Umbria (CEAS), registry No 2695/15 of 16/12/2015.

## Results

We randomly selected a sample of 130 medical charts for each cohort of cases, and 80 medical charts from the cohort of non-cases.

Overall, we retrieved and analysed 767 medical charts for cases: 129 for subarachnoid hemorrhage (ICD-9 430), 125 for intracerebral hemorrhage (ICD-9 431), 129 for other and unspecified intracranial hemorrhage (ICD-9 432.x), 128 for occlusion and stenosis of precerebral arteries (ICD-9 433.x1), 129 for occlusion of cerebral arteries (ICD-9 434.x1), and 127 for acute, but ill-defined, cerebrovascular disease (ICD-9 436) (Table 1).

We retrieved and analysed 78 medical charts for non-cases. Tables 2–7 show the characteristics of the patients for each disease.

As an additional support information file, a minimal anonymized dataset is provided (**S1 Dataset**).

Diagnostic accuracy measures are reported in Table 8.

**Table 1. Characteristics of the overall study sample.**

| ICD-9 codes | Patients N | Sex | | Age | | | Instrumental examinations | | | | Procedures N | Deaths |
|---|---|---|---|---|---|---|---|---|---|---|---|---|
| | | M | F | < 60 | 60–79 | ≥ 80 | CT | MRI | CA | US | | |
| 430 | 129 | 54 | 75 | 35 | 50 | 44 | 124 | 13 | 64 | 0 | 51 | 26 |
| 431 | 125 | 61 | 64 | 17 | 48 | 60 | 124 | 14 | 12 | 0 | 12 | 39 |
| 432 | 129 | 77 | 52 | 12 | 54 | 63 | 127 | 7 | 6 | 0 | 76 | 15 |
| 433.x1 | 128 | 83 | 45 | 6 | 83 | 39 | 91 | 20 | 35 | 0 | 77 | 4 |
| 434.x1 | 129 | 69 | 60 | 18 | 55 | 56 | 127 | 47 | 0 | 77 | 12 | 11 |
| 436 | 127 | 50 | 77 | 5 | 45 | 77 | 118 | 14 | 0 | 43 | 0 | 28 |
| **Overall, N (%)** | **767** | **394 (51)** | **373 (49)** | **93 (12)** | **335 (44)** | **339 (44)** | **711** | **115** | **117** | **120** | **228** | **123 (16)** |

Notes: CT, Computed Tomography; MRI, Magnetic Resonance Imaging; CA, Cerebral Angiography; US, Ultrasound.

## Hemorrhagic stroke

**Subarachnoid hemorrhage.** We identified a cohort of 294 patients with subarachnoid hemorrhage (ICD-9 430), from which we extracted a sample of 130 cases, of these 129 clinical charts were analysed (one clinical chart was not available).

Table 2 shows the basic characteristics of the patients with subarachnoid hemorrhage. The majority of patients were females (58%). Most patients (73%) were > 60 years.

Instrumental examinations performed for the diagnosis included computed tomography (CT) of the head for most patients, followed by arteriography of cerebral arteries, and magnetic resonance (MRI). Forty percent of patients underwent intracranial or neck vessels surgical procedures. Twenty percent of patients died during hospital stay.

**Table 2. Characteristics of patients with subarachnoid hemorrhage who were identified in the Regional Administrative Database of Umbria.**

| Subarachnoid hemorrhage | |
|---|---|
| **Incident cases** (N medical charts reviewed) | 129 |
| **ICD-9 code, N (%)** | |
| 430 Subarachnoid hemorrhage | 129 (100) |
| **Sex, N (%)** | |
| Male | 54 (42) |
| Female | 75 (58) |
| **Age, N (%)** | |
| *< 60* | 35 (27) |
| *60–79* | 50 (39) |
| *≥ 80* | 44 (34) |
| **Instrumental examinations, N (%)** | |
| Computed tomography (CT) of head | 124 (96) |
| Arteriography of cerebral arteries | 64 (50) |
| Magnetic resonance imaging (MRI) of brain | 13 (10) |
| **Surgical procedures, N (%)** | |
| Endovascular repair or occlusion of head and neck vessels | 32 (25) |
| Other surgical occlusion of intracranial vessels | 19 (15) |
| **Deaths, N (%)** | |
| Patients deceased during hospital admission | 26 (20) |

**Table 3. Characteristics of patients with intracerebral hemorrhage who were identified in the Regional Administrative Database of Umbria.**

| Intracerebral hemorrhage | |
| --- | --- |
| **Incident cases** (N medical charts reviewed) | 125 |
| **ICD-9 code, N (%)** | |
| 431 Intracerebral hemorrhage | 125 (100) |
| **Sex** | |
| Male | 61 (49) |
| Female | 64 (51) |
| **Age, N (%)** | |
| < 60 | 17 (14) |
| 60–79 | 48 (38) |
| ≥ 80 | 60 (48) |
| **Instrumental examinations, N (%)** | |
| Computed tomography (CT) of head | 124 (99) |
| Magnetic resonance imaging (MRI) of brain | 14 (11) |
| Arteriography of cerebral arteries | 12 (10) |
| **Procedures, N (%)** | |
| Incision and excision of skull, brain, and cerebral meninges | 6 (5) |
| Ventriculostomy or other operations to establish drainage of ventricle | 6 (5) |
| **Deaths, N (%)** | |
| Patients deceased during hospital admission | 39 (31) |

**Table 4. Characteristics of patients with other and unspecified intracranial hemorrhage who were identified in the Regional Administrative Database of Umbria.**

| Other and unspecified intracranial hemorrhage | |
| --- | --- |
| **Incident cases** (N medical charts reviewed) | 129 |
| **ICD-9 code, N (%)** | |
| 432 Other and unspecified intracranial hemorrhage | 129 (100) |
| 432.0 Nontraumatic extradural hemorrhage | 4 (3) |
| 432.1 Subdural hemorrhage | 121 (94) |
| 432.9 Unspecified intracranial hemorrhage | 4 (3) |
| **Sex, N (%)** | |
| Male | 77 (60) |
| Female | 52 (40) |
| **Age, N (%)** | |
| < 60 | 12 (9) |
| 60–79 | 54 (42) |
| ≥ 80 | 63 (49) |
| **Instrumental examinations, N (%)** | |
| Computed tomography (CT) of head | 127 (98) |
| Magnetic resonance imaging (MRI) of brain | 7 (5) |
| Arteriography of cerebral arteries | 6 (5) |
| **Procedures, N (%)** | |
| Incision of brain and cerebral meninges | 63 (49) |
| Craniotomy or craniectomy | 13 (10) |
| **Deaths, N (%)** | |
| Patients deceased during hospital admission | 15 (12) |

**Table 5. Characteristics of patients with occlusion and stenosis of precerebral arteries who were identified in the Regional Administrative Database of Umbria.**

| Occlusion and stenosis of precerebral arteries | |
|---|---|
| **Incident cases** (N medical charts reviewed) | 128 |
| **ICD-9 code, N (%)** | |
| 433.x1 Occlusion and stenosis of precerebral arteries with cerebral infarction | 128 (100) |
| 433.01 Basilar artery | 3 (2) |
| 433.11 Carotid artery | 119 (93) |
| 433.21 Vertebral artery | 2 (2) |
| 433.31 Multiple and bilateral | 4 (3) |
| 433.81 Other specified precerebral artery | - |
| 433.91 Unspecified precerebral artery | - |
| **Sex** | |
| Male | 83 (65) |
| Female | 45 (35) |
| **Age, N (%)** | |
| < 60 | 6 (5) |
| 60–79 | 83 (65) |
| ≥ 80 | 39 (30) |
| **Instrumental examinations, N (%)** | |
| Computed tomography (CT) of head | 91 (71) |
| Arteriography of cerebral arteries | 35 (27) |
| Magnetic resonance imaging (MRI) of brain | 20 (16) |
| **Procedures, N (%)** | |
| Endarterectomy of other vessels of head and neck | 57 (45) |
| Percutaneous angioplasty or atherectomy of precerebral (extracranial) vessel(s) | 20 (16) |
| **Deaths, N (%)** | |
| Patients deceased during hospital admission | 4 (3) |

**Table 8** shows the diagnostic accuracy measures for subarachnoid hemorrhage: sensitivity 100% (95% CI: 97% - 100%), specificity 96% (95% CI: 90% - 99%), positive predictive value (PPV) 98% (95% CI: 93% - 100%), and negative predictive value (NPV) 100% (95% CI: 95% - 100%). Misclassification of cases and non-cases is described in **Table 9**.

False positives were due to lack of neuroimaging documentation in the medical chart.

**Intracerebral hemorrhage.** We identified a cohort of 1,259 patients with intracerebral hemorrhage (ICD-9 431), from which we extracted a sample of 130 cases, of these 125 clinical charts were analysed (five clinical charts were not available).

**Table 3** shows the basic characteristics of patients with intracerebral hemorrhage. Fifty-one percent of patients were females. Most patients (86%) were >60 years. Almost all patients underwent CT of the head, while about 10% underwent MRI of the brain or arteriography of cerebral arteries (Table 2). Few patients (10%) underwent neurosurgery procedures. Almost one third of patients died during hospitalization.

The diagnostic accuracy measures for intracerebral hemorrhage were: sensitivity 100% (95% CI: 97% - 100%), specificity 98% (95% CI: 91% - 100%), PPV 98% (95% CI: 94% - 100%), and NPV 100% (95% CI: 95% - 100%) **(Table 8)**.

Two patients were considered false positives: one patient had negative instrumental (CT or MRI) diagnosis for intracerebral haemorrhage; in a second patient, imaging confirmation of intracerebral hemorrhage (CT or MRI) could not be found in the medical chart (**Table 9**).

**Table 6. Characteristics of patients with occlusion of cerebral arteries who were identified in the Regional Administrative Database of Umbria.**

| Occlusion of cerebral arteries | |
|---|---|
| **Incident cases** (N medical charts reviewed) | 129 |
| **ICD-9 code, N (%)** | |
| 434.x1 Occlusion of cerebral arteries with cerebral infarction | 129 (100) |
| 434.01 Cerebral thrombosis | 87 (67) |
| 434.11 Cerebral embolism | 24 (19) |
| 434.91 Cerebral artery occlusion, unspecified | 18 (14) |
| **Sex** | |
| Male | 69 (53) |
| Female | 60 (47) |
| **Age, N (%)** | |
| < 60 | 18 (14) |
| 60–79 | 55 (43) |
| ≥ 80 | 56 (43) |
| **Instrumental examinations, N (%)** | |
| Computed tomography (CT) of head | 126 (98) |
| Diagnostic ultrasound of head and neck | 77 (60) |
| Magnetic resonance imaging (MRI) of brain | 47 (36) |
| **Procedures, N (%)** | |
| Injection or infusion of thrombolytic agent | 12 (9) |
| **Deaths, N (%)** | |
| Patients deceased during hospital admission | 11 (9) |

**Other and unspecified intracranial hemorrhage.** We identified a cohort of 419 patients with "other and unspecified intracranial hemorrhage" (ICD-9 432.x), from which we extracted a sample of 130 cases, of these 129 clinical charts were analysed (one clinical chart was not available).

**Table 7. Characteristics of patients with acute, but ill-defined, cerebrovascular disease who were identified in the Regional Administrative Database of Umbria.**

| Acute, but ill-defined, cerebrovascular disease | |
|---|---|
| **Incident cases** (N medical charts reviewed) | 127 |
| **ICD-9 code, N (%)** | |
| 436 Acute, but ill-defined, cerebrovascular disease | 127 (100) |
| **Sex** | |
| Male | 50 (39) |
| Female | 77 (61) |
| **Age, N (%)** | |
| < 60 | 5 (4) |
| 60–79 | 45 (35) |
| ≥ 80 | 77 (61) |
| **Instrumental examinations, N (%)** | |
| Computed tomography (CT) of head | 118 (93) |
| Diagnostic ultrasound of head and neck | 43 (34) |
| Magnetic resonance imaging (MRI) of brain | 14 (11) |
| **Deaths, N (%)** | |
| Patients deceased during hospital admission | 28 (22) |

**Table 8. Cross tabulation of the index test (ICD-9-CM code) for the results of the reference standard (medical chart).**

|  | True Positive | False Positive | True Negative | False Negative |
|---|---|---|---|---|
| 430 Subarachnoid hemorrhage | 126 | 3 | 78 | 0 |
| 431 Intracerebral hemorrhage | 123 | 2 | 78 | 0 |
| 432.x Other and unspecified intracranial hemorrhage | 126 | 3 | 78 | 0 |
| 433.x1 Occlusion and stenosis of precerebral arteries | 89 | 39 | 77 | 1 |
| 434.x1 Occlusion of cerebral arteries | 117 | 12 | 78 | 0 |
| 436 Acute, but ill-defined, cerebrovascular disease | 105 | 22 | 78 | 0 |

**Table 4** shows the basic characteristics of patients with other and unspecified intracranial hemorrhage. The most common ICD-9 subgroup was the code 432.1 (subdural hemorrhage) (94%). Sixty percent of patients were males. Most patients (91%) were 60 years and older. The most frequent diagnostic instrumental examination was CT of the head. Almost half of the patients underwent incision of brain and cerebral meninges.

The diagnostic accuracy measures for other and unspecified intracranial hemorrhage were: sensitivity 100% (95% CI: 97% - 100%), specificity 96% (95% CI: 90% - 99%), PPV 98% (95% CI: 93% - 100%), and NPV 100% (95% CI: 95% - 100%) (**Table 8**).

Three patients were considered false positives: two patients were misclassified because they had a diagnosis of subarachnoid hemorrhage instead of subdural, while one patient had no symptoms or signs in past history (**Table 9**).

## Ischemic stroke

**Precerebral arteries.** We identified a cohort of 468 patients with occlusion and stenosis of precerebral arteries (ICD-9 433.x1), from which we extracted a sample of 130 cases, of these 128 clinical charts were analysed (two clinical charts were not available).

**Table 5** shows the basic characteristics of patients with occlusion and stenosis of precerebral arteries. The most common ICD-9 subgroup was the code 433.11 (carotid artery) (93%). Sixty-five percent of patients were males. Most patients (65%) were in the age class 60–79 years. The most frequent diagnostic instrumental examination was CT of the head. Almost half of the patients (45%) underwent endarterectomy of other vessels of head and neck.

The diagnostic accuracy measures for occlusion and stenosis of precerebral arteries were: sensitivity 99% (95% CI: 94% - 100%), specificity 66% (95% CI: 57% - 75%), PPV 70% (95% CI: 61% - 77%), and NPV 99% (95% CI: 93% - 100%) (**Table 8**).

Thirty-nine patients were considered false positives. Twenty-nine patients who had been discharged with a stroke diagnosis were asymptomatic, admitted for planned carotid endarterectomy, and for most of them (26/29) no instrumental confirmation of the disease was found in the medical chart. Seven patients had no diagnostic report of carotid artery ultrasound in the medical chart. Three patients had carotid stenosis <50% (**Table 10**).

**Table 9. Hemorrhagic stroke: reasons for incorrect identification of cases and controls.**

|  | 430 Subarachnoid hemorrhage | 431 Intracerebral hemorrhage | 432.x Other and unspecified intracranial haemorrhage |
|---|---|---|---|
| **FALSE POSITIVES** | - Instrumental confirmation of subarachnoid hemorrhage (CT or MRI) not found in the medical chart (n. 3). | - Negative instrumental diagnosis for intracerebral hemorrhage (CT or MRI) (n. 1); <br>- Instrumental confirmation of intracerebral hemorrhage (CT or MRI) not found in the medical chart (n. 1). | - Misclassification (diagnosis of subarachnoid hemorrhage instead of subdural) (n. 2); <br>- Symptoms or signs not reported in the medical history (n. 1). |
| **FALSE NEGATIVES** | None | None | None |

**Table 10. Ischemic stroke: reasons for incorrect identification of cases and controls.**

|  | 433.x1 Occlusion and stenosis of precerebral arteries | 434.x1 Occlusion of cerebral arteries | 436 Acute, but ill-defined, cerebrovascular disease |
|---|---|---|---|
| **FALSE POSITIVES** | - **Planned carotid endarterectomy in asymptomatic patients (n.29) with**: *a) instrumental confirmation not found in the medical chart (26) b) negative instrumental diagnosis for cerebral infarction (2) c) negative instrumental diagnosis paracerebral stenosis (1);* - **Carotid artery ultrasound examination not found in the medical chart (n.7) with**: *a) instrumental confirmation for cerebral infarction (3) b) negative instrumental diagnosis for cerebral infarction (4);* - **No evidence of carotid stenosis or moderate or severe stenosis (<50%) (n.3) with:** *a) symptomatic patient with negative instrumental diagnosis for cerebral infarction (2); b) asymptomatic patient with negative instrumental diagnosis for cerebral infarction (1).* | - Negative instrumental diagnosis for cerebral thrombosis or embolism or occlusion of cerebral arteries with cerebral infarction (n. 8); - Acute, but ill-defined, cerebrovascular disease misclassification (n. 3); - Absence of instrumental examinations in the clinical chart (n. 1). | - Symptomatic patients with negative instrumental diagnosis for acute, but ill-defined, cerebrovascular disease (n. 10); - **Symptomatic patients with instrumental confirmation not found in the medical chart (n. 7):** *a) deceased (n. 3, b) > 90 years (n. 3); c) moved from another hospital ward (n. 1)* - Thrombosis with cerebral infarction misclassification (n. 3). - Asymptomatic patients with instrumental confirmation of chronic ischemia (n. 2). |
| **FALSE NEGATIVES** | - Patient with carotid stenosis but without instrumental examination in the medical chart that can confirm the absence of cerebral infarction (n. 1). | None | None |

A less conservative algorithm that considered true positives patients with a planned hospitalization, lead to an increase in specificity from 66% to 86%, and PPV from 70% to 90%.

One patient in the non-case group was considered a false negative because he had carotid stenosis but with instrumental examination missing in the medical chart that can confirm the absence of cerebral infarction.

**Occlusion of cerebral arteries.** We identified a cohort of 4,152 patients with occlusion of cerebral arteries (ICD-9 434.x1), from which we extracted a sample of 130 cases, of these 129 clinical charts were analysed (one clinical chart was not available).

Table 6 shows the basic characteristics of patients with occlusion of cerebral arteries. The most common ICD-9 subgroup was the code 434.01 (Cerebral thrombosis) (67%). Fifty-three percent of patients were males. Most patients (86%) were 60 years and older. Almost all patients (98%) underwent CT of the head. Few patients (9%) underwent pharmacological procedures.

The diagnostic accuracy measures for occlusion of cerebral arteries were: sensitivity 100% (95% CI: 97% - 100%), specificity 87% (95% CI: 78% - 93%), PPV 91% (95% CI: 84% - 95%), and NPV 100% (95% CI: 95% - 100%) (Table 8).

Twelve patients were considered false positives: eight of them had negative instrumental diagnosis for cerebral thrombosis or embolism or occlusion of cerebral arteries with cerebral infarction; three patients were misclassified because they had a diagnosis of acute, but ill-defined, cerebrovascular disease; one patient had no instrumental examinations in the clinical chart (Table 10).

**Acute, but ill-defined, cerebrovascular disease.** We identified a cohort of 278 patients with "acute, ill-defined, cerebrovascular disease" (ICD-9 436), from which we extracted a sample of 130 cases, of these 127 clinical charts were analysed (three clinical charts were not available).

**Table 7** shows the basic characteristics of patients with acute, but ill-defined, cerebrovascular disease. Sixty-one percent of patients were females. Most of patients (61%) were 80 years and older. The most frequent diagnostic instrumental examination was CT of the head (93%). One fifth of the patients died during hospitalization.

The diagnostic accuracy measures for acute, but ill-defined, cerebrovascular disease were: sensitivity 100% (95% CI: 97% - 100%), specificity 78% (95% CI: 69% - 86%), PPV 83% (95% CI: 75% - 89%), and NPV 100% (95% CI: 95% - 100%) (**Table 8**).

Twenty-two patients were considered false positives. Of these, ten were symptomatic patients with negative instrumental diagnosis; seven were symptomatic patients with instrumental confirmation not found in the medical chart; three were misclassified because they had diagnosis of thrombosis with cerebral infarction; lastly, two were asymptomatic patients with instrumental confirmation of chronic ischemia (**Table 10**).

## Discussion

In administrative databases the diagnosis of a given disease is associated with a specific ICD code. Despite some limitations, the ICD code is an important tool designed to map health conditions to corresponding general disease categories, along with specific variations. These codes have the advantage of being widely available and require much lower effort and cost than consulting medical charts [17].

The present study is the first to validate the main ICD-9 codes related to ischemic and hemorrhagic stroke using the Umbria healthcare database. Through the study an excellent level of accuracy for hemorrhagic stroke and acceptable values of specificity and PPV for ischemic stroke was found. This is the second study in Italy to validate ICD-9 codes at regional level. Spolaore et al[18] assessed the accuracy of discharge diagnoses related to stroke criteria and found that the codes 430, 431, 434, and 436 in primary diagnoses had the highest PPVs ranging from 61% to 78%. The authors used the MONICA criteria[19] and obtained similar results to ours in terms of PPVs. Other studies that measured the validity of stroke related ICD-9 codes in Italy were limited to a hospital level[20] [21], or limited to the validity of ischemic stroke only [20] but the results were substantially comparable to those of our study. Potential discrepancies in terms of accuracy might exist between different healthcare databases and these can be explained by the validation criteria used for case ascertainment or inaccuracies in coding the diagnosis. However, validation of administrative databases is context-specific and the results of the present validation study can be generalizable only to the setting of the Umbria population. Hence, the Umbria healthcare database can be used to perform epidemiological and clinical research on health services, in terms of assessment of efficacy, safety and appropriateness of drugs prescription and use of medical devices, through cross-linkage of several databases such as hospital discharge records and prescription databases.

The diagnostic accuracy results of our study are in line with those described in the literature of validation studies using analysis of medical records as the gold standard.

A systematic review published in 2012 [4] identified 35 studies that across 1990–2010 evaluated the validity of algorithms for identifying ischemic and hemorrhagic strokes (intracranial hemorrhage and subarachnoid haemorrhage). All studies included validated administrative coding data in primary and secondary position through abstraction of medical charts. The source population was heterogeneous (i.e. inpatient, outpatient, incident and prevalent cases).

In addition, while two studies validated stroke diagnosis in a paediatric population, most studies conducted the validation process using hospitalization databases, though 16 studies evaluated stroke as a cause-of-death on death certificates and one study reported on outpatient data.

Criteria for the confirmation of stroke varied widely across the studies. More than half of the studies used a specific set of diagnostic criteria based mostly on the WHO criteria (clinical criteria for stroke with CT negative for lesion) to evaluate the stroke diagnosis.

In terms of accuracy, for most studies evaluating codes 430, 431, or 434.x separately, the reported PPVs were 80% or higher. Most of the studies that validated the 436 code had a PPV ≥ 70%. In contrast, most studies reported low PPV values for the 433.x code, with the exception of a study [22] that evaluated the code 433.x1 separately from the code 433.x0 with a much higher value of PPV (71% compared to 13%). Moreover they compared algorithms using the primary discharge diagnosis with those using diagnoses in any position (primary and secondary diagnoses) and found less than 10% higher PPVs for algorithms using the primary discharge diagnosis alone.

A more recent systematic review published in 2015 [5] identified 77 studies that across 1976–2015 evaluated the validity of ICD-9 or ICD-10 codes related to any cerebrovascular disease and found that in more than half of the studies sensitivity was > 82%, specificity > 95% and PPV > 81%. Hospital discharge positions (primary and secondary diagnosis) and type of administrative data (i.e. inpatient, outpatient) were abstracted from the included studies. Most of the studies used chart review as a gold standard. About half of the included studies used chart reviews sometimes in conjunction with unspecific diagnostic criteria as a gold standard, while about 35% used chart review with a specific set of diagnostic criteria, most often the WHO criteria. Finally, about 15% of the studies used regional stroke registers and clinical databases. The review also separately analysed the three codes related to hemorrhagic stroke, while regarding ischemic stroke, in most studies the code ICD-9 436 or ICD-10 I64 was analysed, paired with the ICD-9 434 or ICD-10 I63.

Regarding subarachnoid hemorrhage codes (ICD-9 430 or ICD-10 I60) the PPV was ≥ 86% in 16 of the 26 studies where this was reported. The sensitivity of these codes, reported by 4 studies, ranged from 35% to 95%. Twenty-six studies evaluated the intracerebral hemorrhage codes (ICD-9 431 or ICD-10 I61) and the PPV was ≥ 87% in 16 of the 25 studies reporting on PPV. The sensitivity of these codes, as reported by 3 studies, ranged from 57% to 95%. The codes ICD-9 432 or ICD-10 I62 were less used and evaluated in 15 studies, in which the PPV was ≤ 67% in all but two.

Regarding ischemic stroke the ICD-9 code 433 was evaluated in 19 papers, and in 14 of the 19 the PPV was ≤ 71%. Two studies reported very low measures of sensitivity for ischaemic stroke (ICD-9 433). The code for occlusion of cerebral arteries (ICD-9 434 or ICD-10 I63) had a PPV ≥ 82% in 20 of the 27 studies. The sensitivity, available from 6 papers, ranged from 2% to 80%.

Finally, 16 studies examined the validity of ICD-9 436 and 434 together, and the PPV was ≥ 82% in 10 of the 16.

## Strengths and limitations

A strength of our study is that we used medical charts as reference standards to adjudicate cases of hemorrhagic or ischemic strokes. In addition, we relied on a pre-published protocol with no deviation during the study development. To ensure the quality of reporting we followed the STARD 2015 criteria [16] for diagnostic accuracy studies. Lastly, we used detailed and explicit criteria for case ascertainment, and duplicate and independent processes for medical chart review, data abstraction and analysis. We acknowledge that a potential limitation of our study is that we did not evaluate the accuracy of ICD-9 codes located in secondary position. Another limitation of the present study concerns the generalizability of our results in other populations with different demographic characteristics and disease prevalence.

Nonetheless, the study methodology could be replicated in other settings to identify possible differences in diagnostic accuracy results.

## Conclusion

The Regional Health Authority of Umbria has started a research activity in the last years regarding case definitions of several diseases [23–26], and has validated ICD-9 codes for several oncological diseases [27–30]. In the present study, we validated the ICD-9 diagnostic codes for important cerebrovascular diseases using the Regional Health database of Umbria. Findings from the present study suggest that the assessed ICD-9 codes are highly predictive of hemorrhagic stroke while they may be considered acceptable for ischemic stroke. In conclusion, our results showed that the Umbrian healthcare administrative database, validated for these diseases, can be confidently used for epidemiological, outcome, and health services research.

## Supporting information

**S1 Table. STARD-2015-Checklist.**
(DOCX)

**S1 Dataset. Minimal anonymized dataset.**
(PDF)

## Author Contributions

**Conceptualization:** Massimiliano Orso, Francesco Cozzolino, Alessandro Montedori, Giuseppe Ambrosio, Iosief Abraha.

**Data curation:** Massimiliano Orso, Francesco Cozzolino, Iosief Abraha.

**Formal analysis:** Massimiliano Orso, Francesco Cozzolino, Paolo Eusebi, Iosief Abraha.

**Funding acquisition:** Alessandro Montedori, Giuseppe Ambrosio.

**Investigation:** Massimiliano Orso, Francesco Cozzolino, Anna Mengoni, Iosief Abraha.

**Methodology:** Massimiliano Orso, Francesco Cozzolino, Iosief Abraha.

**Project administration:** Alessandro Montedori, Giuseppe Ambrosio, Iosief Abraha.

**Resources:** Marcello De Giorgi, David Franchini, Anna Julia Heymann, Alessandro Montedori, Iosief Abraha.

**Software:** Alessandro Montedori, Giuseppe Ambrosio.

**Supervision:** Alessandro Montedori, Giuseppe Ambrosio, Iosief Abraha.

**Validation:** Massimiliano Orso, Francesco Cozzolino, Serena Amici, Alessandro Montedori, Iosief Abraha.

**Visualization:** Massimiliano Orso, Francesco Cozzolino, Anna Julia Heymann, Iosief Abraha.

**Writing – original draft:** Massimiliano Orso, Francesco Cozzolino, Giuseppe Ambrosio, Iosief Abraha.

**Writing – review & editing:** Massimiliano Orso, Francesco Cozzolino, Serena Amici, Marcello De Giorgi, David Franchini, Paolo Eusebi, Anna Julia Heymann, Guido Lombardo, Anna Mengoni, Alessandro Montedori, Giuseppe Ambrosio, Iosief Abraha.

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
