## [Decision Letter · Decision Letter 0]

6 Nov 2019

PONE-D-19-24032

Validity of cerebrovascular ICD-9-CM codes in healthcare administrative databases. The Umbria Data-Value Project

PLOS ONE

Dear Dr. Abraha,

Thank you for submitting your manuscript to PLOS ONE. After careful consideration, we feel that it has merit but does not fully meet PLOS ONE’s publication criteria as it currently stands. Therefore, we invite you to submit a revised version of the manuscript that addresses the points raised during the review process.

This study is well conducted. The authors should highlight what is their incremental contribution in terms of methodology to previous studies.

We would appreciate receiving your revised manuscript by Dec 21 2019 11:59PM. To enhance the reproducibility of your results, we recommend that if applicable you deposit your laboratory protocols in protocols.io, where a protocol can be assigned its own identifier (DOI) such that it can be cited independently in the future. For instructions see: http://journals.plos.org/plosone/s/submission-guidelines#loc-laboratory-protocols

We look forward to receiving your revised manuscript.

Kind regards,

Gianni Virgili

Academic Editor

PLOS ONE

Journal Requirements:

2. During our internal evaluation of the manuscript, we found significant text overlap between your submission and the following previously published works, on which you are an author:

https://journals.plos.org/plosone/article?id=10.1371/journal.pone.0218919

We recognize that the publication cited above was written by you and/or your co-authors. However, please note that re-use of text from a previous publication is unacceptable according to PLOS ONE’s editorial policy on text overlap and re-use (http://journals.plos.org/plosone/s/ethical-publishing-practice#loc-plagiarism). Although this previously published article is Open Access, we still ask that you acknowledge the reuse of any text or data, citing your previous article, so as to properly attribute the original published source. We thank you for your attention to our editorial policies.

Additional Editor Comments (as further peer-review):

The manuscript is very clearly written. The study is well designed with an appropriate choice of controls and adequate validation methods.

The Discussion presents details on previous large systematic reviews on this topic, which encompass a larger time span and include studies using different methodologies. One of these reviews also provides subgroups analyses, which are also presented. The authors should highlight what is the incremental value of their methodology with respect to previous research that could be used in future research and in practice.

As a minor comment, avoid presenting numbers (accuracy estimates) in the Abstract conclusions.

Reviewers' comments:

Reviewer's Responses to Questions

**Comments to the Author**

1. Is the manuscript technically sound, and do the data support the conclusions?

Reviewer #1: Yes

2. Has the statistical analysis been performed appropriately and rigorously? 

Reviewer #1: Yes

3. Have the authors made all data underlying the findings in their manuscript fully available?

Reviewer #1: Yes

4. Is the manuscript presented in an intelligible fashion and written in standard English?

Reviewer #1: Yes

5. Review Comments to the Author

Reviewer #1: This research article focused on validation of ICD-9 codes for several cerebrovascular conditions in the Umbria regional administrative database in order to use them for future epidemiological studies.

The following MAJOR ISSUES should be carefully considered by the authors:

1) The Authors created several group of diagnoses and demographic and clinical characteristics of patients were described. I suggest adding a new table with all cases in order to describe characteristics of general cohort.

2) In the discussion section authors reported what is already available in the literature. However, I miss what this specific study adds to the topic. I suggest to the authors to stress the key-message of the study also in the discussion section.

3) The authors declared that they focus on ICD9 diagnosis in primary position. What about the same diagnosis in the second position? Are difference in the number of cases and validity of codes?

6. PLOS authors have the option to publish the peer review history of their article (what does this mean?). If published, this will include your full peer review and any attached files.

Reviewer #1: No

---

## [Author Response · Author response to Decision Letter 0]

9 Dec 2019

Da: PLOS ONE <em@editorialmanager.com>

A: Iosief Abraha <iosief_a@yahoo.it>

Inviato: mercoledì 6 novembre 2019, 10:59:46 CET

Oggetto: PLOS ONE Decision: Revision required [PONE-D-19-24032] - [EMID:1be9306ff293a376]

PONE-D-19-24032

Validity of cerebrovascular ICD-9-CM codes in healthcare administrative databases. The Umbria Data-Value Project

PLOS ONE

Dear Dr. Abraha,

Thank you for submitting your manuscript to PLOS ONE. After careful consideration, we feel that it has merit but does not fully meet PLOS ONE’s publication criteria as it currently stands. Therefore, we invite you to submit a revised version of the manuscript that addresses the points raised during the review process.

This study is well conducted. The authors should highlight what is their incremental contribution in terms of methodology to previous studies.

We would appreciate receiving your revised manuscript by Dec 21 2019 11:59PM. To enhance the reproducibility of your results, we recommend that if applicable you deposit your laboratory protocols in protocols.io, where a protocol can be assigned its own identifier (DOI) such that it can be cited independently in the future. For instructions see: http://journals.plos.org/plosone/s/submission-guidelines#loc-laboratory-protocols

A rebuttal letter that responds to each point raised by the academic editor and reviewer(s). This letter should be uploaded as separate file and labeled 'Response to Reviewers'.

A marked-up copy of your manuscript that highlights changes made to the original version. This file should be uploaded as separate file and labeled 'Revised Manuscript with Track Changes'.

An unmarked version of your revised paper without tracked changes. This file should be uploaded as separate file and labeled 'Manuscript'.

We look forward to receiving your revised manuscript.

Kind regards,

Gianni Virgili

Academic Editor

PLOS ONE

Journal Requirements:

2. During our internal evaluation of the manuscript, we found significant text overlap between your submission and the following previously published works, on which you are an author:

https://journals.plos.org/plosone/article?id=10.1371/journal.pone.0218919

We recognize that the publication cited above was written by you and/or your co-authors. However, please note that re-use of text from a previous publication is unacceptable according to PLOS ONE’s editorial policy on text overlap and re-use (http://journals.plos.org/plosone/s/ethical-publishing-practice#loc-plagiarism). Although this previously published article is Open Access, we still ask that you acknowledge the reuse of any text or data, citing your previous article, so as to properly attribute the original published source. We thank you for your attention to our editorial policies.

Authors’ reply: We modified most of the text overlapping with our previous publication.

Additional Editor Comments (as further peer-review):

The manuscript is very clearly written. The study is well designed with an appropriate choice of controls and adequate validation methods.

Authors’ reply: We thank the Editor for this comment.

The Discussion presents details on previous large systematic reviews on this topic, which encompass a larger time span and include studies using different methodologies. One of these reviews also provides subgroups analyses, which are also presented. The authors should highlight what is the incremental value of their methodology with respect to previous research that could be used in future research and in practice.

Authors’ reply: We added a paragraph in the Discussion section describing what our study adds to the existing literature in the field, with a focus on a comparison with other Italian validation studies.

As a minor comment, avoid presenting numbers (accuracy estimates) in the Abstract conclusions.

Authors’ reply: Thanks for your suggestion. We have deleted the accuracy estimates from the Abstract conclusions.

Reviewers' comments:

Reviewer's Responses to Questions

Comments to the Author

1. Is the manuscript technically sound, and do the data support the conclusions?

Reviewer #1: Yes

2. Has the statistical analysis been performed appropriately and rigorously?

Reviewer #1: Yes

3. Have the authors made all data underlying the findings in their manuscript fully available?

Reviewer #1: Yes

4. Is the manuscript presented in an intelligible fashion and written in standard English?

Reviewer #1: Yes

5. Review Comments to the Author

Reviewer #1: This research article focused on validation of ICD-9 codes for several cerebrovascular conditions in the Umbria regional administrative database in order to use them for future epidemiological studies.

The following MAJOR ISSUES should be carefully considered by the authors:

1) The Authors created several group of diagnoses and demographic and clinical characteristics of patients were described. I suggest adding a new table with all cases in order to describe characteristics of general cohort.

Authors’ reply: We added a new table (Table 1) with a summary of the characteristics of the overall sample. 

2) In the discussion section authors reported what is already available in the literature. However, I miss what this specific study adds to the topic. I suggest to the authors to stress the key-message of the study also in the discussion section.

Authors’ reply: We added a paragraph in the Discussion section describing what our study adds to the existing literature in the field, with a focus on a comparison with other Italian validation studies.

3) The authors declared that they focus on ICD9 diagnosis in primary position. What about the same diagnosis in the second position? Are difference in the number of cases and validity of codes?

Authors’ reply: We decided to limit our validation study to only consider the codes in primary position because, according to the Italian legislation, the primary diagnosis constitutes the main cause of the need for treatment and/or diagnostic tests, and is mainly responsible for the use of resources.

We acknowledged this in the study limitations: “We acknowledge that a potential limitation of our study is that we did not evaluate the accuracy of ICD-9 codes located in secondary position”. 

6. PLOS authors have the option to publish the peer review history of their article (what does this mean?). If published, this will include your full peer review and any attached files.

Do you want your identity to be public for this peer review? For information about this choice, including consent withdrawal, please see our Privacy Policy.

Reviewer #1: No

---

## [Editor Report · Decision Letter 1]

26 Dec 2019

Validity of cerebrovascular ICD-9-CM codes in healthcare administrative databases. The Umbria Data-Value Project

PONE-D-19-24032R1

Dear Dr. Abraha,

We are pleased to inform you that your manuscript has been judged scientifically suitable for publication and will be formally accepted for publication once it complies with all outstanding technical requirements.

With kind regards,

Gianni Virgili

Academic Editor

PLOS ONE
---

## [Editor Report · Acceptance letter]

31 Dec 2019

PONE-D-19-24032R1 

Validity of cerebrovascular ICD-9-CM codes in healthcare administrative databases. The Umbria Data-Value Project 

Dear Dr. Abraha:

I am pleased to inform you that your manuscript has been deemed suitable for publication in PLOS ONE. Congratulations! Your manuscript is now with our production department. 

With kind regards,

on behalf of

Dr. Gianni Virgili 

Academic Editor

PLOS ONE